# Phenotypes of Floral Nectaries in Developmental Mutants of Legumes and What They May Tell about Genetic Control of Nectary Formation

**DOI:** 10.3390/biology11101530

**Published:** 2022-10-19

**Authors:** Andrey Sinjushin

**Affiliations:** Department of Genetics, Faculty of Biology, Lomonosov Moscow State University, Leninskie Gory 1-12, 119234 Moscow, Russia; asinjushin@mail.ru or sinjushin@mail.bio.msu.ru

**Keywords:** androecium, corolla, dorsalization, evolution, monosymmetry

## Abstract

**Simple Summary:**

The third largest angiosperm family, Leguminosae, is remarkable with the outstanding diversity of its flowers, usually monosymmetric and adapted to different pollination strategies. A key attractant of leguminous flowers is nectar. Compared with *Arabidopsis* (Brassicaceae), very little is known about regulation of floral nectaries development in legumes. This work aimed to investigate details of these nectaries’ morphology in flowers of mutants of different legume species. It was found that the changes in identity of petals and stamens usually do not affect a proper structure and position of nectaries in leguminous flowers, thus suggesting a high stability of attracting structures versus the pronounced plasticity of perianth and stamens. Some of genes involved in regulation of nectary development in *Arabidopsis* seem to have the same functions in legumes. The principal difference between *Arabidopsis* and legumes is connected with a flower monosymmetry in most representatives of the latter taxon, which is also reflected in structure of their floral nectaries.

**Abstract:**

The vast majority of angiosperms attracts animal pollinators with the nectar secreted through specialized floral nectaries (FNs). Although there is evidence that principal patterns of regulation of FN development are conserved in large angiosperm clades, these structures are very diverse considering their morphology and position within a flower. Most data on genetic control of FN formation were obtained in surveys of a model plant species, *Arabidopsis thaliana* (Brassicaceae). There are almost no data on genetic factors affecting FN development in Leguminosae, the plant family of a high agricultural value and possessing outstandingly diverse flowers. In this work, the morphology of FNs was examined in a set of leguminous species, both wild-type and developmental mutants, by the means of a scanning electron microscopy. Unlike Brassicaceae, FNs in legumes are localized between stamens and a carpel instead of being associated with a certain floral organ. FNs were found stable in most cases of mutants when perianth and/or androecium morphology was affected. However, regulation of FN development by *BLADE-ON-PETIOLE*-like genes seems to be a shared feature between legumes (at least *Pisum*) and *Arabidopsis*. In some legumes, the adaxial developmental program (most probably *CYCLOIDEA*-mediated) suppresses the FN development. The obtained results neither confirm the role of orthologues of *UNUSUAL FLORAL ORGANS* and *LEAFY* in FN development in legumes nor reject it, as two studied pea mutants were homozygous at the weakest alleles of the corresponding loci and possessed FNs similar to those of wild-type.

## 1. Introduction

Adaptations of flowering plants to interaction with different animal pollinators include evolution of attractants, such as perianth shape and color, floral scent, and secretion of nectar or other edible substances. The observed diversity of flowers among extant angiosperms (at least in cases where flowers can be unambiguously distinguished from inflorescences) accords with an outstanding variety of floral nectaries (FNs) [1]. Here and further, FNs are defined as floral structures producing sacchariferous secrete regardless of their position, homology, and mode of secretion.

The regulatory pathways controlling development of FNs are far from being precisely dissected. In a model plant species, *Arabidopsis thaliana* (L.) Heynh. (Brassicaceae), FNs comprise small glands bearing secretory stomata and localized at the outer bases of all six stamens (extrastaminal FNs). As indicated in studies of floral mutants of *A. thaliana*, development of FNs does not require a proper differentiation of stamens themselves [2]. If mutations cause homeotic transformation of stamens into carpels or petals, FNs nevertheless emerge. Development of the FNs is restricted to the third floral whorl, which is androecial in wild-type flowers [2]. Later, it was found that proper differentiation of some floral whorls is critical for FN development (see [3] for review).

A key regulator *CRABS CLAW* (*CRC*) was reported to control both carpel closure and FN development in *A. thaliana* [4]. After a survey involving a wider range of taxa it was concluded that *CRC* most probably was recruited for regulation of FNs development near the base of the eudicots, as its expression was observed in FNs of several eudicot families but not of Ranunculaceae [5]. Surprisingly, *CRC* is expressed even in the extrafloral nectaries, i.e., those localized outside flower, although their position and morphology are quite diverse even among eudicots [5].

Additional factors which control FNs development in *A. thaliana* are two *BLADE-ON-PETIOLE* genes (*BOP1* and *BOP2*), both expressed in FNs [6]. In flowers of double mutants *bop1 bop2*, only small bulges lacking secretory stomata develop at the sites of normal FNs location. However, CRC expression is retained in *bop1 bop2* mutants, suggesting that *BOP1/BOP2* and *CRC* act independently in regulation of FNs development [6]. Many more genes are expected to be involved in FNs regulation, including those directly interacting with *CRC*, as ca. 120 transcription factors exhibited ability to bind with *CRC* promoter in experiment ([7], see also papers cited therein).

The third largest angiosperm family, Leguminosae, includes numerous representatives of high agricultural value, as well as convenient model species, such as *Pisum sativum* L., *Medicago truncatula* Gaertn., and *Lotus japonicus* (Regel) K. Larsen. For both reasons, numerous leguminous mutants with anomalies in flower development are known (e.g., [8]). As compared with Brassicaceae, flowers are much more diverse in Leguminosae with respect to their symmetry, merism, synorganization between different domains, as well as different patterns of multiplication and reduction of various floral parts [9]. Although not many legumes have been characterized regarding structure of their FNs, studies in three aforementioned model genera indicate that their FNs are of different morphology than those of *Arabidopsis* [10,11,12]. In monosymmetric flowers of these genera, secretory parts of FNs comprise areas of stomata in the abaxial part of receptacle and/or hypanthium between stamens and carpel (intrastaminal FNs). These areas can be somewhat elevated, but their borders are not clearly distinct from the adjacent parts of receptacle/hypanthium (Figure 1C). In some legumes, FNs may be clearly elevated above the receptacle level; such FNs seem associated with a carpel base, not with staminal whorl (Figure 2C). In both Leguminosae (at least most of them) and Brassicaceae, nectar is released through modified stomata.

Gene *CRC* seemingly retains its role in regulation of FNs development in leguminous flowers. Its expression was found in sites of FN localization in both *Pisum* and *Medicago* [13]. Unfortunately, in available literature there are no descriptions of secretory pattern and FN morphology in floral mutants or transgenic forms of legumes. ‘Double’ (having no stamens and carpels but producing numerous petals) flowers of ornamental form of *Lotus corniculatus* were reported to produce no nectar and were not visited by insects [14]. However, the regulation of FN development and activity in legumes is a matter of significant interest also from practical point of view, as many legumes are valuable melliferous plants (*Medicago*, *Melilotus*, *Onobrychis*, *Robinia* and many others). Teuber et al. [11] provided evidence for efficiency of selection for higher nectar productivity in *Medicago*. In addition, a seed set in cross-pollinating species is dependent on visitation by insects.

From fundamental point of view, data on FN regulation obtained for *A. thaliana* can hardly be entirely approximated to the Leguminosae with their monosymmetric and outstandingly diverse flowers. In this connection, it is of interest to widen the existing view on genetic control of FN development in legumes. This work aims to investigate structure of FNs in floral mutants of several leguminous species and uncover possible participation of different genes (or, more broadly, regulatory pathways) in FN development.

## 2. Materials and Methods

The accessions which served as material for a given survey are listed in Table 1.

**Table 1 biology-11-01530-t001:** Plant material used for the study.

Genotype	Accession	Orthologous Gene in *Arabidopsis*	Floral Phenotype of Mutant	Reference
*Pisum sativum* L.				
*bivexillum* (*biv*)	JI3056	Unknown	Adaxial sepals petaloid, sometimes no petals, stamens free, their number reduced (Figure 1Q)	[15]
*cochleata* (*coch*)	JI2758 (nonsense), Wt11304 (missense)	*BLADE-ON-PETIOLE**1*, *2*	Varying from reduction of stamen number and production of extra flag instead of keel to impairment of staminal and keel fusion (Figure 1S,U)	[16]
*keeled wings* (*k*)	F_2_ JI2163 × WL1749, F_2_ ‘Chlorophyll-13’ × WL1238	*PsCYC2* *	Wings similar to keel petals (Figure 1F)	[17]
*stamina pistilloida* (*stp-1*)	JI2163, F_2_ JI2163 × ‘Cheburashka’	*UNUSUAL FLOWER ORGANS*	Outer adaxial stamens converted into carpels, petals with sepaloid sectors (Figure 1M and Appendix A)	[18]
*superpetaloidum* (*sup*)	F_2_ JI1340 × ‘Cheburashka’	Unknown	Outer (usually adaxial) stamens petaloid (Figure 1K)	[19]
*stp-1 sup*	F_2_ JI2163 × ‘Cheburashka’	–	Outer adaxial stamens produce petaloid and carpelloid excrescences (Figure 1O)	–
*unifoliata*-tendrilled acacia (*uni*^tac^)	Az-23	*LEAFY*	Keel petals free, adnate to stamens; abaxial stamens fused together (Figure 1H,I)	[20]
Unknown	cv. Anvend	Unknown	Wings symmetric (Figure 1D)	–
Wild-type	F_2_ JI2163 × ‘Cheburashka’, F_2_ JI2163 × WL1749, F_2_ ‘Chlorophyll-13’ × WL1238	–	–	–
*Cajanus cajan* (L.) Millsp.				
*partial cleistogamy* (*pct*)	ICPB 2203	Unknown	Wing and keel petals symmetric, keel free, stamens free (Figure 2D)	[21]
*lanceolate* (*llt*)	ICP 5529	Unknown	Wings almost symmetric, keel petals free and narrow (Figure 2F)	[22]
Wild-type	ICPL 20325	–	–	–
*Wisteria* spp.				
*W. floribunda* f. *violaceoplena* (C.K. Schneid.) Rehder & E.H. Wilson	Ornamental (Turkey)	Unknown	Calyx widely splayed, petals numerous and deformed, stamens petaloid or rarely fertile, carpel deformed (Figure 3B and Appendix A)	[23]
*W. sinensis* (Sims) Sweet, wild-type	Living collection of the Tsitsin Main Botanical Garden, Moscow, Russia	–	–	–
*Clitoria ternatea* L.				
*C. ternatea* var. *pleniflora* Fantz	Ornamental (Thailand)	Unknown	All five petals flag-like (Figure 4E), stamens free	–
Wild-type		–	–	–
*Caragana arborescens* Lam.				
Wild-type	Ornamental (Russia)	–	–	–
*C. arborescens* f. *lorbergii* Koehne	Living collections of the Tsitsin Main Botanical Garden and the Schroeder Arboretum of the Moscow Timiryazev Agricultural Academy, Moscow, Russia	Unknown	Wings and keel petals slightly narrowed, flag very narrow, keel often unfused, carpel incompletely sealed (Appendix A)	[24]
*Calpurnia aurea* (Aiton) Benth.				
Wild-growing	Ethiopia; herbarium specimens (MW): MW0583527, MW0583528	–	–	–

* Orthologue of *CYCLOIDEA* first described in *Antirrhinum majus* L. (Plantaginaceae). Dash = not applicable.

**Figure 3 biology-11-01530-f003:**
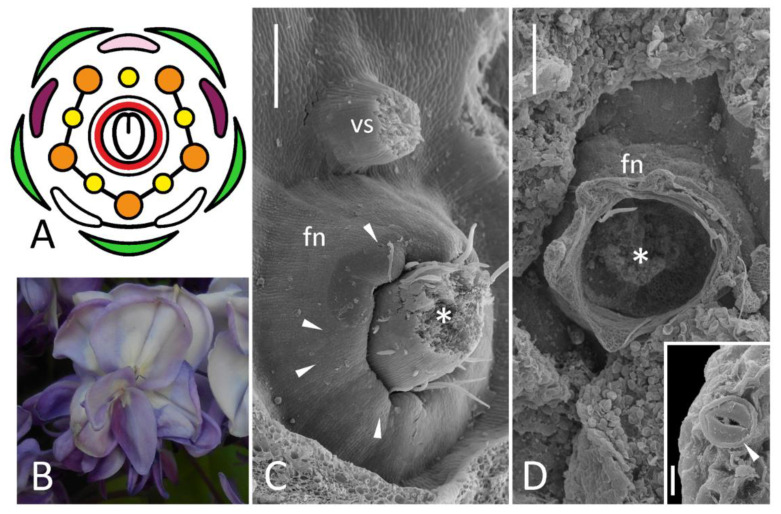
Floral diagram (**A**) and position of FN (**C**) of wild-type flower of *Wisteria sinensis* compared with an overall morphology (**B**) and position of FN (**D**) of ‘double’ flower of *W. floribunda* f. *violaceoplena*. Key: arrowheads = exemplary nectariferous stomata (enlarged in inset of (**D**); asterisk = carpel base or place where it was attached; fn = floral nectary; vs = vexillary stamen. For color designations on floral diagram, see Figure 1. Scale bars: 300 μm (**C**,**D**), 10 μm (**D**, inset).

**Figure 4 biology-11-01530-f004:**
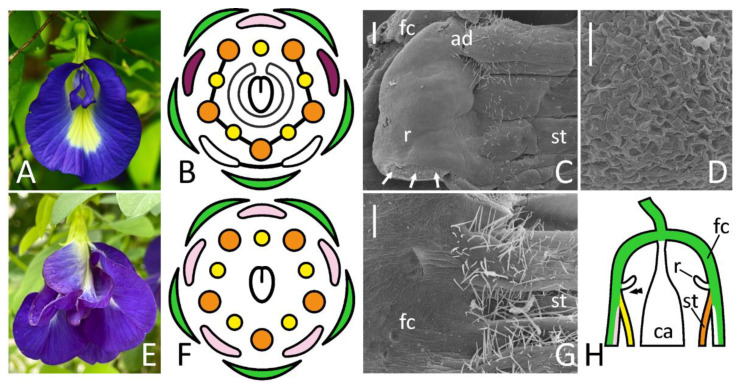
Floral morphology (**A**,**E**), floral diagrams (**B**,**F**) and SEM images of identical areas on a border between floral cup and androecium (**C**,**D**,**G**) in wild-type (**A**–**D**) and ‘double’ (**E**–**G**) flowers of *Clitoria ternatea*. The position of illustrated area in a resupinated flower is schematically represented on (**H**). A marginal part of ridge-like structure of wild-type flower is enlarged on (**D**). Key: arrows = rupture caused by the dissection; double arrowhead = place illustrated in (**C**,**G**); ad = adaxial side; ca = carpel; fc = floral cup; r = ridge; st = stamens [note absence of staminal fusion on (**F**)]. For color designations on floral diagrams and (**H**), see Figure 1. Scale bars: 300 μm (**C**,**G**), 30 μm (**D**). Photos: Arturo C. Mendoza (**A**), Janardhan Uppada (**E**).

Flowers of *Cajanus* accessions were collected on the experimental plot of the International Crops Research Institute for Semi-Arid Tropics (ICRISAT, Hyderabad, India). Anomalous flowers of *Wisteria* were sampled in the living collection of the Nezahat Gökyiğit Botanical Garden (Istanbul, Turkey), while normal flowers were sampled from the specimen in the living collection of the Tsitsin Main Botanical Garden (Moscow, Russia). Both forms of *Clitoria* were found as freely growing in different provinces of Thailand, most probably escaping from gardens. All accessions of a garden pea were grown on experimental plot at the Skadovskii Zvenigorod Biological Station (Moscow region, Russia), except for coch lines (JI2758 and Wt11304). These were grown in the open field plot and in the phytotron (21 ± 1.5 °C, 16 h of light) at the All-Russian Research Institute of Agricultural Microbiology (Leningrad region, Russia).

A minimum of three (3–11) flowers were examined for each genotype, preferentially from different individual plants. Freshly collected flowers were fixed in 70% ethanol, dissected under a stereomicroscope and prepared for electron microscopy as described in [25]. In the case of *Calpurnia* herbarium specimen, desiccated flowers were soaked in hot water (90–95 °C), thermostated at 60 °C in 70% ethanol for 24 h and then stored in 70% ethanol at room temperature. After this, the material was dissected and prepared for SEM.

Some images were captured with an Olympus SZ61 stereomicroscope (Olympus, Tokyo, Japan) equipped with a UHCCD05000KPA camera (ToupTek Photonics, Hangzhou, China). Digital images were processed for publication with Corel PHOTO-PAINT 2017 (Corel Corporation, Ottawa, ON, Canada). As corolla phenotypes of many floral mutants were already described and illustrated in previous publications (Table 1), only the most significant features of flower macromorphology were represented in figures of this work.

As studied accessions of the same species were not isogenic, they were compared on qualitative level, i.e., whether FN is present or absent. The presence of FN was judged by the occurrence of stomata.

## 3. Results

### 3.1. Overall Floral Morphology

All studied species belong to the subfamily Papilionoideae and normally possess the so-called papilionate corolla (‘flag blossom’) with three discernible petal types, viz. flag (adaxial), two wings (lateral) and a keel composed of two fused abaxial petals (Figure 1A,B, Figure 2A, Figure 4A, Figure 5A and Figure 6A). Ten stamens are arranged in two whorls with nine of them fusing into incomplete tube (Figure 1B). A filament of the inner adaxial stamen opposed to a flag petal (vexillary stamen) is typically free from adjacent filaments (diadelphous androecium) or secondarily reunites with them (pseudomonadelphous androecium). In both cases, there may be slits along a free stamen’s filament and/or two fenestrae at its base providing access to FN. In flowers of *C. aurea*, stamens are fused only with their bases, but a vexillary stamen is adnate only to one of adjacent stamens, so a slit is asymmetrically open from one side. To judge whether certain mutations distorted FN morphology or not, it was important to describe FN structure in wild-type flowers of all examined taxa. Despite an overall similarity of corolla and androecium, FNs appeared quite diverse in a selected set of species, which is reasonable to expect considering that these species are not closely related.

### 3.2. Floral Nectaries

#### 3.2.1. *Pisum sativum*

In agreement with a previous description [10], FN in pea normally comprises a crescent-shaped elevation bearing numerous secretory stomata (Figure 1C). Some minor differences were found between accessions and even between flowers collected from the same accession, considering the exact number of nectariferous stomata or shape of the elevated part of FN.

FNs of typical morphology were found in all mutants with a normal number of organs regardless of whether these organs were of normal morphology. Recombinants homozygous for *k* (with keel-like wings) and cv. Anvend (with symmetric wings, Figure 1D) possessed FNs similar to those of wild-type plants (Figure 1D,F). FNs of *k* recombinants were the largest among all examined pea accessions and expanded till the adaxial side of a carpel from both sides (Figure 1F,G).

FNs were found normal in mutants having stamens transformed into carpels (*stp-1*, Figure 1N), petals (*sup*, Figure 1L) or sharing features of three organs in double mutants *stp-1 sup* (Figure 1P). Such replacement usually affected two adaxial outer stamens (but not free, i.e., vexillary, stamen between them), while the abaxial domain together with FN remained intact. In late flowers of *stp-1* with more pronounced floral anomalies, numerous stomata were observed on the abaxial sides of ectopic carpels’ bases (Appendix A). These stomata differ from those of FN with their sizes and morphology. This phenomenon needs a further examination.

Two examined *coch* accessions exhibited either weak (Wt11304) or severe (JI2758) floral phenotypes (Table 1). In both, the expression of anomaly was variable. Some (although not all) flowers of JI2758 possessed two opposed flags, one abaxial and one adaxial, while fewer other floral parts often developed than in wild-type flowers (Figure 1S; see also Figure 3H,J in [16]). In Wt11304, petals were of atypical shape, keel petals were free and stamens fused irregularly (Figure 1U), but the overall number of floral organs was usually the same as in wild-type flowers. In flowers of this accession, FNs were present at their proper sites (Figure 1V), while no FNs were found in flowers of JI2758 (Figure 1T). It was true not only for flowers with two opposed flags, but also for those with more or less proper differentiation of five petals.

The accession JI3056 registered as a type line for mutation *biv* [26] appeared phenotypically heterogenous, probably comprising F_2_ population from some *biv* × *BIV* cross, i.e., it included both mutant and wild-type plants, which also differed in stem length and leaf morphology. A range of floral anomalies was variable between individual mutant plants and even between different flowers of the same plant. In severe cases, two adaxial sepals were flag-like in their shape and pigmentation, while corolla consisted of fewer petals or sometimes was completely reduced (Figure 1Q). Fewer stamens developed together with chimeric stamen-petal structures, fusion between stamens was also aberrant. Such anomalous flowers usually produced no seeds, as their carpels were often deformed and contained no ovules, although a primary description of this mutant reported its fertility [15]. In mild cases, corolla included symmetric wings and free keel petals, while fusion of stamens was partly distorted. The number of secretory stomata was strikingly reduced in some flowers of *biv* plants (Figure 1R).

In *uni*^tac^ plants, a certain (lesser) fraction of flowers developed with anomalies, such as aberrations of keel fusion and amalgamation of two or three abaxial stamens into one, so the resulting number of stamens was eight or nine instead of ten (Figure 1I). However, despite all these changes that affected the abaxial domain, FN remained normal in such flowers (Figure 1J).

#### 3.2.2. *Cajanus cajan*

Although a pigeon pea is a crop of high importance in tropical latitudes, data on structure and ontogeny of its FNs are almost missing. That is why it is worth describing the wild-type morphology of FN in this plant species.

In (pre)anthetic flower, FN is a circular elevation surrounding a carpel base with a more or less pronounced depression from the adaxial part, probably resulting from the pressure of the vexillary stamen (Figure 2C). In the abaxial part, this circular rim is somewhat higher than in the adaxial sector. Numerous nectariferous stomata develop along FN margin and on its inner and, partly, outer surfaces (Figure 2C). These stomata are distributed unevenly. On the adaxial side of FN, they are much less abundant than on the abaxial side.

In both examined floral mutants of a pigeon pea, FNs are present (Figure 2G,I). Their shape differs between accession (for example, FN has five lobes in *pct* mutant, Figure 2I–K). It is unclear whether these features accompany floral mutations or result from other genotypic differences. The compared lines are not isogenic: for example, mutant *pct* appeared in F_2_ progeny from interspecific cross *C. cajan* × *C. lineatus* (Wight & Arn.) Maesen [21]. However, one may state confidently that these two mutations do not arrest FN development.

#### 3.2.3. *Wisteria* spp.

Normal flowers of *W. sinensis* bear a clearly discernible annular FN surrounding a carpel base with stomata along its rim, on its inner surface and, rarely, on its outer slope (Figure 3C).

‘Double’ flowers of *W. floribunda* have all organs typical for wild-type, but their structure is modified. A widely splayed calyx encloses numerous petals, some of which are differentiated as counterparts of a typical papilionate corolla, while others are deformed and not identifiable (Figure 3B). Stamens are also petaloid, sometimes connate with their filaments. Surprisingly, there is a deformed carpel in a center of a receptacle (Appendix A) surrounded with a collar-like elevation. Only fully open flowers at late anthesis were available for examination and this collar was partly decayed, but there were sparse stomata along its margin evidencing for its secretory function (Figure 3D). It indicates that FN is present in ‘double’ flowers of *Wisteria*.

#### 3.2.4. *Clitoria ternatea*

The first available description of a putative FN in *C. mariana* L., a species of the same genus, was given as early as in 1879 [27]. The data for *C. ternatea* presented here agree with this primary report in that flower of this genus contains an incomplete annular ridge interrupted at its adaxial side (Figure 4B,C). Whereas Trelease [27] recognized this ridge as FN, there are no discernible stomata on either its side (Figure 4B and Appendix A), which suggests that it does not serve as secretory structure. Its margin bears cells with concave outer surfaces, some of which collapse (Figure 4D). This may point at some kind of secretory activity (such as scent-producing osmophore). Alternatively, one may suggest that nectar is released through cell walls rather than through modified stomata in *Clitoria*, which is not typical for legumes. Mature flowers of *Clitoria* are resupinate, i.e., their flag is oriented downwards (Figure 4A), so probably this ridge prevents nectar, if any, from outflow. This ridge’s orientation is pocket- or valve-like (Figure 4C,H), so this hypothesis is reasonable.

No signs of this ridge were found in an ornamental form of this species with all petals flag-like (Figure 4E,G).

#### 3.2.5. *Caragana arborescens*

Wild-type flowers of *C. arborescens* produce numerous nectariferous stomata at the abaxial part of receptacle lacking a discernible elevation (Figure 5C), which agrees with the earlier description [25].

*Caragana arborescens* f. *lorbergii* is remarkable with narrow petals (especially flag) and female sterility, which is most probably associated with an incomplete carpel closure (Appendix A). However, stomata are present at the adaxial part of receptacle (Figure 5D,E) and, in some flowers, even laterally (Figure 5D).

#### 3.2.6. *Calpurnia aurea*

*Calpurnia* with its typically papilionate corolla (Figure 6A) is relatively closely related to *Cadia* possessing atypical polysymmetric flowers (see below). For this reason, floral buds of *C. aurea* were included in analysis. Only herbarium specimen of a wild-growing plant was available for examination, so cells of FN were shrunken partly obscuring its shape (Figure 6B). FN in this plant represents a ring around a stipe base at the bottom of pronounced hypanthium (Figure 6B). As distinct from other studied species, secretory stomata are present on all circumference of FN ring, so it is completely polysymmetric (Figure 6B).

## 4. Discussion

### 4.1. BOP-Mediated Regulation of Nectary Development Is Conserved in Pisum

To date, *BOP*-like genes and their mutant alleles have been discovered and investigated at least in four legume species, *P. sativum*, *M. truncatula* [16], *L. japonicus* [28], and *Lupinus angustifolius* L. [29]. The cited researches focused primarily on role of these genes in regulation of nitrogen-fixing symbiotic nodule formation, the aspect which cannot be studied in *Arabidopsis*. It was demonstrated that *BOP*-like genes in legumes participate in control of development of flower and leaf (especially stipules), as well as in organ abscission [29]. However, no information on FNs was reported in mutants at corresponding loci.

The *Ljnbcl1* (*noot-bop-coch-like1*) mutants of *L. japonicus* have no extrafloral nectaries normally placed at the leaf base [28], while nothing is known about their FNs. However, in *Lotus* these extrafloral nectaries are commonly interpreted as stipules-derived (see references in [28]). All *coch* mutants of pea have reduced or completely absent stipules at lower nodes [16]. Hence it is not surprising that orthologous genes control development of stipules (as well as their derivatives) in two legume species.

The results reported here indicate that FNs are missing in flowers of a pea line JI2758 (Figure 1T) homozygous for deletion allele of gene *COCH*, a pea orthologue of *BOP1*/*BOP2*. The interpretation of floral phenotype of *coch* mutants is still pending, but anomalous flowers may have extra organs in all whorls (suggesting a ‘loss of determinacy’: [30]) and/or features of ‘dorsalization’, i.e., production of two opposed flags in both adaxial and abaxial floral domains (see Figure 3J in [16]). While a pea flower normally bears FN only at its abaxial domain (Figure 1C), it could be suggested that anomalous ‘dorsalization’ causes the expansion of an adaxial patterning and hence the loss of abaxial identity. However, FNs are absent even in flowers with a proper abaxial-adaxial differentiation of petals, which can be also found in line JI2758. It evidences that *COCH*, a pea orthologue of *BOP1*/*BOP2* genes, is also responsible for proper development of FNs, which is probably a conserved function of these genes in angiosperms (or at least rosids including both Brassicaceae and Leguminosae).

In flowers of line Wt11304, which is a homozygote at missense mutation [16], FNs are present (Figure 1V). It agrees with the overall mildness of phenotypic manifestation of mutation in Wt11304.

### 4.2. Flower Dorsalization May Inhibit Development of Floral Nectaries, but Not Necessarily

Floral morphology of the ornamental ‘double’ form of *Clitoria* can be interpreted as resulting from dorsalization, i.e., expansion of adaxial developmental pattern to all domains. All petals become flag-like, so all antepetalous stamens are differentiated as vexillary and remain unfused. In such flower, annular ridge is completely lost (Figure 4E) and no putative secretory stomata can be found within a floral cup. As circular ridge is normally absent from the adaxial side, it is not surprising that adaxialization of the whole circumference of developing floral meristem associates with a complete suppression of ridge (and probably FN) development.

As evidenced from studies of model legume species with monosymmetric flowers [17,31], TCP genes (such as orthologues of *CYCLOIDEA* first described in *Antirrhinum majus*) play a decisive role in diversification of petal types. Although the completely dorsalized phenotype was only hypothesized by Wang et al. [17] in the Leguminosae, there is a single confirmed case, when naturally occurring morphology results from such dorsalization. It is the papilionoid genus *Cadia* with all five petals of a polysymmetric flower controlled by a *CYC*-like gene, which is normally expressed in the adaxial domain [32]. Such mechanism of evolutionary shift in floral symmetry cannot be excluded in the other lineages. However, as seen from the case of dorsalized flowers of *Clitoria*, such homeotic scenario of floral evolution associates with a risk of FN loss (Figure 4D, 6B). Nectary structure is not documented in *Cadia*, but there is evidence that its flowers produce abundant nectar as an adaptation to bird pollination [33]. Stamens of *Cadia* bear knob-like excrescences on filament bases, which Cronk and Ojeda [33] identified as ‘nectar globes’, i.e., visual cues attracting pollinators. These globes are very unlikely secretory themselves: there are no visible stomata on them, as judged from Figure 34 in [34]. Moreover, there are no records of filament nectaries in the Leguminosae. In addition to visual attraction, these ‘globes’ most probably prevent nectar outflow from a campanulate flower lacking nectar-collecting cavity (such as spur) and/or protect nectar from consumption by inefficient visitors, such as insects.

It is of significant interest to examine FN morphology in *Cadia* to learn how the corolla dorsalization may combine with FN maintenance. No material of this plant was available for study, but molecular phylogenetic studies recurrently supported a close relationship between *Cadia* and *Calpurnia* (see [35] and references cited therein). The examination of *Calpurnia* flowers indicated that their nectar-secreting stomata are distributed more or less equally on a hypanthium (Figure 6B), i.e., there is no suppression of their emergence on adaxial side. Although close relation between *Cadia* and *Calpurnia* does not readily guarantee similarity of their FNs, these observations indicate that even highly pronounced floral monosymmetry may be compatible with a polysymmetric FN. In such cases, the dorsalization (i.e., expansion of adaxial regulatory pattern to all circumference of floral meristem) would not remove FN (Figure 6D). Following Linnean terms, peloria (anomalously polysymmetric flowers of normally monosymmetric taxa) can be either nectariferous (*peloria nectaria*, as *Cadia*) or nectarless (*peloria anectaria*, as *C. ternatea* var. *pleniflora*) [36] (and references cited therein). The innovations changing floral symmetry and simultaneously removing FNs either would be inadaptive or require changes in pollination strategy.

As seen from the results reported here, a similar perianth structure (flag blossom) associates with different FN morphology. A deeper examination of spatial pattern of expression of *CYC*-like genes is required in papilionoid flowers. Most probably, the expression of *CYC* orthologues may either spread to the adaxial sector of area between stamens and carpel (thus inhibiting adaxial FN development, Figure 6C) or not (thus not preventing development of a polysymmetric FN, Figure 6D). As an alternative hypothesis, the adaxial markers may express early to establish a perianth monosymmetry, but then their expression weakens and has no decisive role in suppression of development of FNs, which, in their turn, initiate relatively late.

Whereas *CYC*-like genes are preferentially expressed adaxially and serve as markers of morphological and functional adaxiality, they are most probably also act as suppressors of FN development in monosymmetric flowers of certain taxa. Indeed, in legumes, FNs are usually placed either uniformly around a carpel base or abaxially, which correlates with a flower (mono)symmetry [25,37]. However, there is at least one case of a monosymmetric leguminous flower in which an adaxial FN position was reported, *Lespedeza* [38]. In *Flemingia*, secretory stomata occupy two distinct fields in abaxial and adaxial floral domain (Sinjushin, unpublished). FNs are localized exclusively on the adaxial side of monosymmetric flowers in different angiosperm families, such as Lythraceae (*Cuphea*), Cleomaceae (*Cleome*, *Polanisia*) or Polygalaceae (*Salomonia*). The adaxial placement of FN in the latter family is especially remarkable, as Polygalaceae and Leguminosae belong to the same order, Fabales. FNs are of different morphology, position (outside of staminal whorl(s) or between stamens and carpel) and probably origin in listed families. It means that suppression of FN development from the adaxial side (most probably *CYC*-mediated) is not a common rule for monosymmetric flowers for all angiosperms, nor even for Leguminosae. Oppositely, in some taxa FNs are specific for the adaxial floral domain.

### 4.3. Production of Extra Petals Does Not Prevent FN Development

In *A. thaliana*, the best known mutation causing a ‘double’ flower phenotype is *agamous* (*ag*) [39]. Flowers of *ag* mutants develop FNs [2], but genes of the C-lineage including *AG* are required for activation of *CRC*, as evident from phenotypes of double and triple mutants of *A. thaliana* [3], as well as from surveys in other angiosperm taxa [40].

To date, ‘double’ flowers are known in several legume species, such as *P. sativum* (*petalosus* (*pe*) mutants: [8]), *L. corniculatus* [14], *Genista tinctoria* (this species normally has no FNs), and *W. floribunda*. The latter species was examined in the given research. In ‘double’-flowering *L. corniculatus*, no nectar secretion was reported [14], while in *W. floribunda* ring-like FN is found even in ‘double’ flowers (Figure 3D). There is no evidence that ‘double’ floral phenotype in *Wisteria* is connected with *ag*-related mutation. However, ‘double’ flowers of *Arabidopsis* and *Wisteria* have distinct morphologies. In *ag* mutants, after inception of sepals and petals, a floral meristem produces petals instead of stamens and then continues its proliferation thus giving rise to an ‘infinite’ flower consisting of numerous sepals, petals and chimeric organs [39]. In *W. floribunda* f. *violaceoplena*, floral meristem terminates with a carpel, as in wild-type flowers (Figure 3D). As ontogenetic observations of *Wisteria* indicate, soon after calyx initiation, carpel primordium emerges together with primordia of antesepalous stamens and ‘common’ alternisepalous primordia, the latter developing into petals and antepetalous stamens [41]. Most probably, a ‘double’ floral phenotype in *Wisteria* is connected with an excessive proliferation of common primordia rather than of floral apex itself, so flower bears a carpel and FN is patterned at its base. The *pe* mutant of pea couples anomalous proliferation of floral apex (additional floral organs are found inside an unfused carpel) and subdivision of common primordia [8]; there is no data on FNs in *pe* flowers.

### 4.4. Can Data from Arabidopsis Be Approximated to Legumes?

Brassicaceae and Leguminosae belong to different orders (Brassicales and Fabales, respectively) within the same clade of rosids [42] and share many common principles of floral developmental regulation, although some differences exist [8], primarily connected with diverse floral symmetries (Figure 7). While in *A. thaliana* FNs are associated with third whorl organs on structural and regulatory levels [2], it is not the case of leguminous flowers, where FNs seem either unbound with any floral structures (*Pisum*: Figure 1C; *Caragana*: Figure 5B) or associated with a carpel (*Cajanus*, *Wisteria*: Figure 2C and Figure 3C). No FNs emerge at bases of ectopic stamen-derived carpels in *stp**-1* mutants of pea (not shown).

The data reported previously [13] indicate that *CRC* most probably remains a key regulator or, at any rate, a marker of FN development in Leguminosae. At least one feature of regulatory similarity between legumes (*Pisum*) and *A. thaliana* is that the activity of *BOP*-like gene(s) is essential for FN development (Figure 1T and Figure 7).

Most of the mutants examined in a given work are not characterized on molecular level. In two pea mutants, *stp-1* and *uni*^tac^, causative mutations are associated with orthologues of *UNUSUAL FLORAL ORGANS* (*UFO*) and *LEAFY* (*LFY*) of *A. thaliana*, respectively [18,20]. Both *LFY* and *UFO* control FN development in *A. thaliana*: FNs are aberrant in single mutants *lfy* and *ufo*, while no FNs develop in double mutants *lfy ufo* [2]. However, FNs of pea mutants *uni*^tac^ and *stp-1* are of normal morphology (Figure 1J,N). Both recessive alleles are the weakest mutations of the respective loci [18,20], which may explain their mild effect on floral morphology. The presence of FNs in these two mutants is not sufficient to conclude that orthologues of *LFY* and *UFO* are not involved in regulation of FN development in pea.

The most notable difference in the control of FN ontogeny between *A. thaliana* and legumes is connected with an adaxial suppression of FN development, most probably *CYC*-mediated (see above). The involvement of *CYC* orthologues in such negative regulation can be hypothesized not only from the phenotype of ‘dorsalized’ forms of *Clitoria* (Figure 4E), but also from the remarkable expansion of FN to lateral domains in *k* mutants of pea (Figure 1G). Gene *K* is one of three known pea orthologues of *CYC*, viz. *PsCYC2* [17]. Normally in pea FN develops only in the abaxial part (Figure 1C), so their expansion to lateral parts in *k* mutants may be due to loss of negative control from *PsCYC2*.

The causative mutation in *C. arborescens* f. *lorbergii* has not been identified yet. This ornamental form with its very narrow leaflets and petals (especially flag) as well as female sterility is reminiscent of legume mutants with defects associated with WUSCHEL-related homeobox1 (WOX1) transcription factor, such as *lathyroides* of pea [43]. The expansion of nectariferous area beyond the abaxial floral domain (Figure 5D) may probably result from a weaker expression of petals identity genes, such as *CYC* orthologues. However, the nature of mutation needs to be revealed to judge more reasonably about the exact mechanisms.

Although basic principles of control of FN ontogeny may be conserved between legumes and *A. thaliana*, the observations reported here evidence for certain taxon-specific differences in this control (Figure 7).

## 5. Conclusions

The ancestral floral groundplan of the Leguminosae (pentamerous and pentacyclic monosymmetric flower) changes significantly in different lineages of this outstandingly diverse family [9]. Evolutionary innovations such as polyandry, multicarpelly, fusion of different floral parts, homeosis, reduction of petals and/or stamens in certain positions, as well as transition to polysymmetry, arose independently in many clades [9]. However, the nectar remains the principal attractant for pollinators. As the observations reported here indicate, FNs in leguminous flowers seem nonresponsive to most of the changes which occur to the outwards of their position, i.e., in perianth and androecium. This provides a basis for significant plasticity of floral morphology without a risk for fitness. Oppositely, among known developmental leguminous mutants, FNs are lost/impaired as a part of complex syndromes negatively affecting numerous floral and extrafloral structures, sometimes causing sterility, e.g., in *coch* or *biv* mutants. Such mutations are rather deleterious and are probably of little or no value for floral evolution.

## Figures and Tables

**Figure 1 biology-11-01530-f001:**
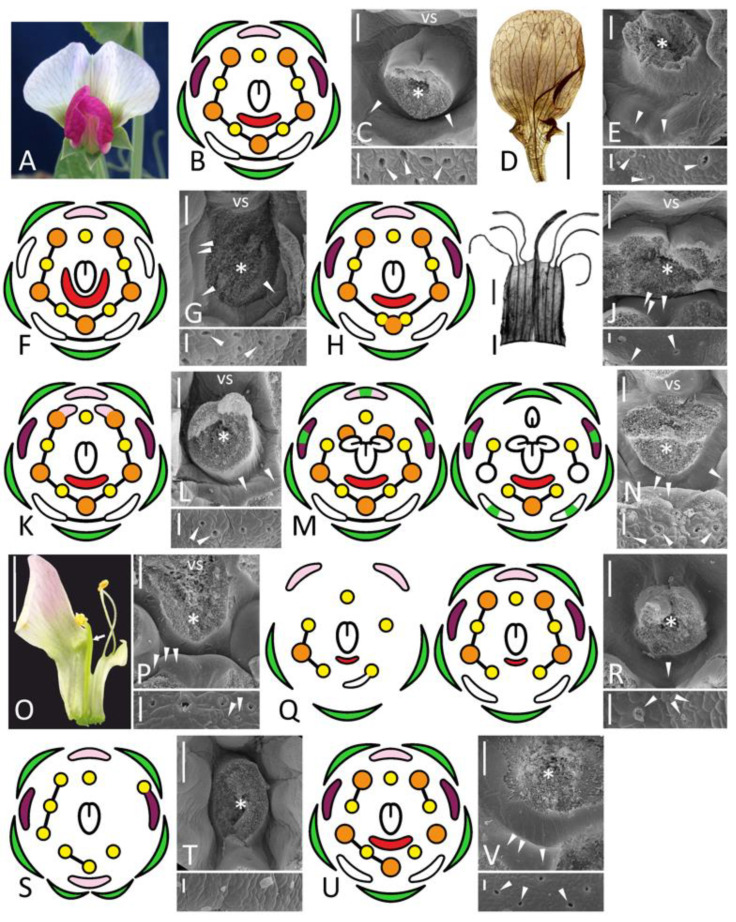
Morphology of flowers and FNs in studied accessions of *Pisum sativum* represented by floral diagrams (**B**,**F**,**H**,**K**,**M**,**Q**,**S**,**U**), SEM images (**C**,**E**,**G**,**J**,**L**,**N**,**P**,**R**,**T**,**V**), and details of macromorphology (**A**,**D**,**I**,**O**). In each SEM image, the upper part represents an overall top view, while the lower part is a closer view of secretory area or a similar region in nectarless flower (**T**). All SEM images are oriented with an abaxial side downwards. (**A**–**C**), wild-type flower of a recombinant from F_2_ JI2163 × WL1749; (**D**,**E**), cv. Anvend with unusual symmetric wings (**D**); (**F**,**G**), *k* recombinant; (**H**–**J**), *uni*^tac^ mutant [in (**I**), vexillary stamen is not shown]; (**K**,**L**), *sup* recombinant; (**M**,**N**), *stp-1* mutants with mild (**M**, left) and severe (**M**, right) phenotype; (**O**,**P**), *stp-1 sup* recombinant; (**Q**,**R**), *biv* recombinant [in (**Q**), left diagram represents severe floral phenotype, while right diagram is a weak phenotype]; (**S**,**T**), severe *coch* phenotype (JI2758; all stamens are represented as inner, as their position was unclear); (**U**,**V**), weak *coch* phenotype (Wt11304). Key: green color = sepal; pink color = flag; purple color = wing; white color = keel; red color = secretory area; orange color = outer stamen; yellow color = inner stamen; line between organs on diagram = fusion (not shown in a calyx); arrowheads = exemplary nectariferous stomata; asterisk = carpel base or place where it was attached; vs = vexillary stamen. Scale bar: 1 mm (**D**,**O**), 0.2 mm (**I**), 300 μm (upper parts of all SEM images), 30 μm (lower parts of all SEM images). Photos: Polina Mamoshina (**A**), Elina Shnayder (**O**).

**Figure 2 biology-11-01530-f002:**
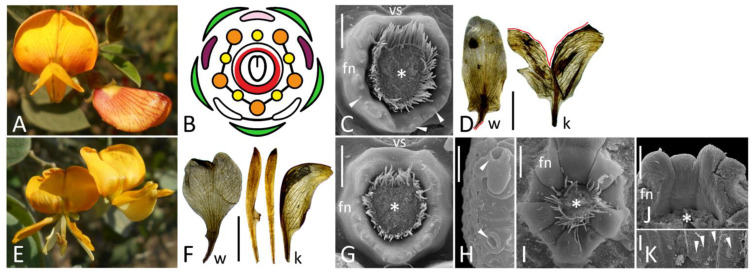
General view (**A**,**E**), diagram (**B**), petal morphology (ethanol-fixed material: **D**,**F**), and details of FN morphology (**C**,**G**–**K**) of flowers of different accessions of *Cajanus cajan*. (**A**–**C**), wild-type flower; (**E**–**H**), *llt* mutant; (**D**,**I**–**K**), *pct* mutant. (**H**) is an enlarged portion of (**G**); (**K**) is an enlarged portion of (**J**), while (**J**) is the same FN as on (**I**) but seen from inside after removal of carpel base. (**C**,**G**,**I**) are oriented with their abaxial side downwards. Key: red line (**D**) = petal damage caused by the dissection; arrowheads = exemplary nectariferous stomata; asterisk = carpel base or place where it was attached; fn = floral nectary; k = keel petal [diversity within a single accession is visible on (**E**,**F**)]; vs = vexillary stamen; w = wing. For color designations on floral diagram, see Figure 1. Scale bars: 0.5 cm (**D**,**F**), 300 μm (**C**,**G**,**I**,**J**), 30 μm (**H**,**K**).

**Figure 5 biology-11-01530-f005:**
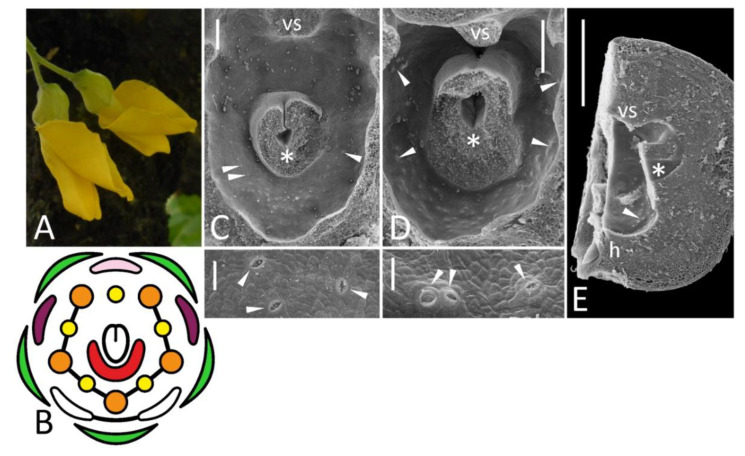
Floral morphology (**A**, photo; **B**, diagram) and localization of FNs (**C**–**E**, SEM images) in normal *Caragana arborescens* (**A**–**C**) and *C. arborescens* f. *lorbergii*: (**D**), top view, (**E**), longitudinal section. In (**C**,**D**), the upper part represents an overall view, while the lower part is a closer view of secretory area. All SEM images are oriented with an abaxial side downwards. Key: arrowheads = exemplary nectariferous stomata; asterisk = carpel base or place where it was attached; h = hypanthium; vs = vexillary stamen. For color designations on floral diagrams, see Figure 1. Scale bar: 1 mm (**E**), 300 μm (upper parts of **C**,**D**), 30 μm (lower parts of **C**,**D**).

**Figure 6 biology-11-01530-f006:**
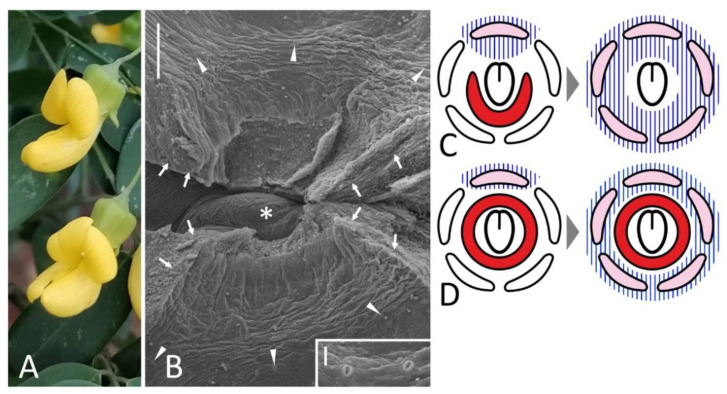
Monosymmetric flowers (**A**) with a polysymmetric FN (**B**) in *Calpurnia aurea* with exemplary stomata shown in inset and hypothetical consequences of dorsalization in flower with (**C**) and without (**D**) adaxial suppression of FN development. Key: arrows = rupture caused by the dissection; arrowheads = exemplary nectariferous stomata; asterisk = former carpel position; blue hatching = adaxial developmental program; pink color = petal of abaxial identity; red color = secretory area. Scale bars: 300 μm (**B**), 30 μm (inset). Photo: Muhammad Adamjee (**A**).

**Figure 7 biology-11-01530-f007:**
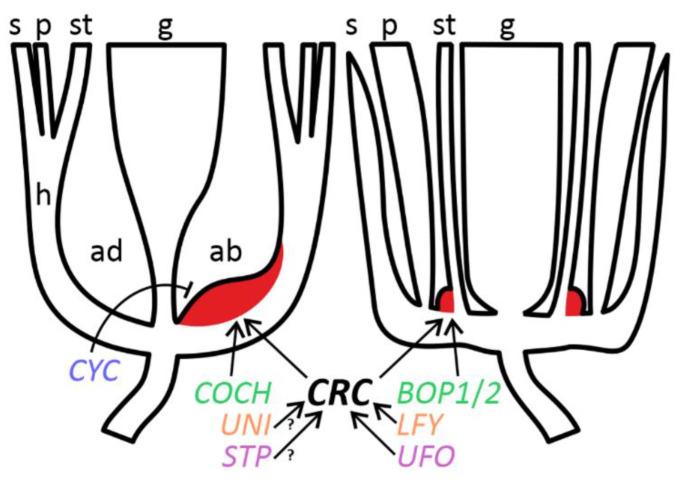
Schematic representation of floral morphology and regulation of FN development comparing *Pisum* (**left**) and *Arabidopsis* (**right**). Orthologous genes are designated with the same color. Key: arrow = positive regulation; blunt arrow = negative regulation; red color = FN; ab = abaxial floral domain; ad = adaxial floral domain; g = gynoecium; h = hypanthium; p = petals; s = sepals; st = stamens; ? = hypothetical pathway.

## Data Availability

Not applicable.

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
