# Peer review of "Phenotypes of Floral Nectaries in Developmental Mutants of Legumes and What They May Tell about Genetic Control of Nectary Formation"

_biology, 2022, doi:10.3390/biology11101530_

Round 1
Reviewer 1 Report (Previous Reviewer 1)
The author carefully considered all previously proposed alterations in the text and revised the paper making the necessary improvements. The title was brought more in accordance with the content and an additional picture was included as suggested. I would like to note since Mendel the mechanisms of genetic control have not changed, but our understanding and investigation methodology of these processes are now deeper and more precise than several decades ago.
At its present version the paper can be published in the Journal.
With best regards,
Author Response
Dear colleague,
Many thanks for your high estimate of my work and positive decision about its acceptability for publication. I am especially glad that the process of review and revision became a kind of true discussion rather than a formal procedure.
Thank you again, I wish you all the best.
Best regards,
Andrey Sinjushin
Reviewer 2 Report (New Reviewer)
The manuscript discusses the variants of genetic control of the floral nectaries in some legume species and their mutants. It is gratifying that such a study did appear not for Arabidopsis, but for a significantly different material.
The manuscript is structured logically. Everything that is stated in the title and in the introduction is described and discussed almost without repetition. But a simple Summary, Abstract and Conclusions are written in very different ways. Of these, the Abstract is the most informative and better reflects the content of the study.
The introduction lacks a phrase about what the author will call floral nectaries in this study? Since nectaries, including floral ones, are diverse. For example see: Fahn A (1979) Secretory tissues in plants. Academic Press, New York - Benouaiche P (1979) Ultrastructure, development and secretion in the nectary of banana flowers. Ann Bot 44:85-93; or here Nepi, M. (2007). Nectary structure and ultrastructure. In: Nicolson, S.W., Nepi, M., Pacini, E. (eds) Nectaries and Nectar. Springer, Dordrecht. https://doi.org/10.1007/978-1-4020-5937-7_3. In the introduction, a few words should be devoted to this issue.
It should be written that the author will judge nectaries by the presence or absence of nectary stomata.
The illustrations are very good diagrams. They really help to figure out what you need to see in these VERY small photographs. It would be nice to have a larger photo.
Which genes do not affect the presence or absence of nectaries is very well outlined in the discussion. But in the Conclusions and a simple Summary, only very generalized thoughts are stated. In the Abstract only BLADE-ON-PETIOLE and CYCLOIDEA are mentioned, and those that do not affect are not mentioned. It's a pity, because this is also the merit of this work.
See other comments in pdf file.

Author Response
See attached file.

This manuscript is a resubmission of an earlier submission. The following is a list of the peer review reports and author responses from that submission.
Round 1
Reviewer 1 Report
The manuscript entitled "Genetic control of flower nectary formation in legumes based on developmental mutant data" describes the identification of morphological changes in flower nectaries of developmental mutants of the Leguminosae family.
Unfortunately, I believe that the manuscript does not meet the Journal standards at the present form and requires some important revision. The article is more descriptive than experimental.
General comments:
- First of all, authors are recommended to change the title, because genetic control involves the study of the genes themselves and their manifestations, and not only the analysis of the morphological manifestations of these mutations. The title is misleading about the meaning of the article.
- The discussion section is consistently laid out and built logically. However, it is necessary to insert at least a simplified diagram of the interaction of the genes referred to in the article (for example, based on a similar network diagram for Arabidopsis, or mutant analysis https://www.ncbi.nlm.nih.gov/pmc/articles/PMC3244948/pdf /tab.0127.pdf/?tool=EBI or
Slavkovic, F.; Dogimont, C.; Morin, H.; Boualem, A.; Bendahmane, A. The genetic control of nectary development. Trends Plant 579 Sci. 2021, 26, 260–271. ets.). It is necessary to reflect the difference in families or missing paths.
- The conclusions that the author makes are very controversial: “Oppositely, mutations which impair a proper development of FNs are associated with complex changes in floral morphology and are most probably of little or no value for floral evolution".
It is not clear on what basis the author made such conclusions. Changes in the structure of the flower can directly affect the formation of seeds, especially with cross-pollination (this can be leveled if the plant changes its reproduction strategy, for example, switches to apomixis). And undoubtedly it can influence the prosperity of the species in the future.
- Language is clear and not requiring a deep revision, I highlighted in yellow some parts that need to be changed.
With great regards

Reviewer 2 Report
My comments are as follows,
1. What scientific question does the authors want to reveal? The diversity of the FN in Legumes? or Genetic Control of Floral Nectaries Formation in Legumes?
2. It is recommended to add color photos of each species investigated in this study.
3. The author mentioned that numerous nectariferous stomata appeared on FN, is the floral nectary stable in morphology or structure within the genus? such as the stomata number, as well as the density of it.
4. The discussion should focus on the diversity and differentiation of mature nectaries in legumes.
5. Some correlation analysis of FN and floral structure in these species should be added, because it is very interesting that the location of floral nectaries in the family are the same, but their size and elevation degree are different among species: a crescent-shaped elevation in Pisum sativum, annular shape in Cajanus cajan and Wisteria, an incomplete annular ridge but does not serve as secretory structure in Clitoria ternatea, and a receptacle lacking a discernible elevation in Caragana arborescens, a polysymmetric FN in a monosymmetric flower of Calpurnia aurea.
